# Assessment of Condition Diagnosis System for Axles with Ferrous Particle Sensor

**DOI:** 10.3390/ma16041426

**Published:** 2023-02-08

**Authors:** Sung-Ho Hong, Hong-Gyu Jeon

**Affiliations:** 1Department of Mechanical System Engineering, School of Creative Convergence Engineering, Dongguk University-WISE Campus, Gyeongju 38066, Republic of Korea; 2Test and Assessment Technology Team, Hyundai Construction Equipment, Yongin 16891, Republic of Korea

**Keywords:** axle, condition diagnosis, ferrous particle sensor, construction equipment

## Abstract

This study presents a condition diagnosis system based on a ferrous particle sensor to estimate the durability of axles in construction equipment. Axles are mechanical devices that play the role of the differential gear in construction equipment that move with wheels and require high reliability. In the durability testing of new axles, failure identification and real-time diagnosis are required. One of the typical failure modes of an axle is increased ferrous-wear particles due to metal-to-metal contact. Therefore, a condition diagnostic program based on the ferrous particle sensor is developed and applied in the bench tests of axles. This program provides information on the amount of wear with respect to ferrous particles using a simple diagnostic algorithm. Additionally, it allows separation and storage of measured data that exceed the reference values; the system provides warnings using color, sound, and pop-up windows to facilitate diagnosis. In the two tests, the first case detected a failure, but in the other case, the sensor did not detect it even though a failure occurred. From the results of bench tests, it is confirmed that the sensor location is a critical factor. Therefore, a multi-physics-based analysis method is suggested for positioning the ferrous particle sensor.

## 1. Introduction

Maintenance is essential for mechanical systems to ensure that they continue to perform their desired functions. The key objective of maintenance is to deploy the minimum amount of resources to achieve maximum reliability from the equipment with the least possible breakdown time and cost [1]. Maintenance is broadly classified into two groups: reactive and proactive. Reactive maintenance is also called breakdown maintenance and results in unscheduled downtime as well as unplanned production losses. Conversely, preventive and predictive maintenance are types of proactive maintenance. Preventive maintenance involves time-based repairs or replacement of components at regular intervals. However, this results in the non-utilization of the full operating lives of the components [2], for which a better alternative is predictive or condition-based maintenance. Predictive maintenance uses data analysis tools and techniques to detect anomalies and possible defects in the equipment such that they can be rectified before failure. To ensure good maintenance of the equipment, condition monitoring is applied in various mechanical systems, which reduces downtime and boosts production efficiency. Among several condition monitoring methods, there are two types of monitoring that were broadly applied in the past; compliance monitoring to diagnose machine conditions based on reference values of physical quantities, such as pressure or temperature; structure integrity monitoring to diagnose machine conditions by measuring the stress and strain of a structure with a strain gauge. 

Nowadays, in addition to these methods, other techniques are widely used, such as diagnosing the machine state using vibration and noise, thermography, lubricant analysis, and non-destructive techniques such as ultrasound [3].

Construction equipment has a high risk of failure owing to its operation in harsh environments. Therefore, it is necessary to reduce failure rates and downtime via condition monitoring. However, traditional condition monitoring methods have problems, such as the low frequency of monitoring, inefficient reporting methods, and low quality of manually collected data [4]. Among the various condition monitoring methods, we focus on condition monitoring based on oil analysis. Figure 1 shows the oil analysis services of three representative construction equipment companies. Caterpillar has performed scheduled oil sampling (S.O.S) fluid analysis since 1971; Volvo has also provided oil analysis services through oil sampling; Hitachi has applied an oil monitoring system containing an integrated oil sensor. Moreover, several construction equipment companies have conducted research on oil sensor-based condition diagnosis systems.

Several lubricants are used in construction equipment. Hence, various oil sensors suitable for measuring lubricant properties such as viscosity, acid number, and wear particles, have been developed. In the present study, a condition monitoring system based on an oil sensor was developed for the axles of construction equipment. On wheeled construction equipment, axles have functions similar to the differential gears in cars, and they are composed of several gears and bearings. Wear particles are generated due to metal-to-metal contact. In axles, most of them show adhesive wear, abrasive wear, and fatigue wear. So, a condition diagnosis system comprising a ferrous particle sensor is needed for diagnosing abnormalities. 

Wear particle sensors commonly utilize inductance and capacitance-based methods [5,6,7,8,9,10,11,12,13,14,15,16,17,18,19,20,21], acoustic methods based on ultrasonic transducers [22,23], optical methods [24,25,26], and a method based on a combination of a permanent magnet and inductance [27], as shown in Figure 2. Sensors for measuring wear particles are being developed in various ways, but most of them have been developed based on inductance and capacitance. Besides oil condition diagnosis based on wear particle sensors, oil condition diagnosis using a complicated sensor that measures various properties at once with one sensor is also applied [3,28]. 

Wear is a phenomenon closely related to machine condition, maintenance, and reliability. Therefore, it is important to determine the amount or type of wear [29]. It is also important to identify the particles generated by wear in lubricating systems. 

Wear particles typically include ferrous, non-ferrous, and ceramic debris. Of these, the ferrous particle comprises the largest portion because many machine components are made of steel; most components of axles are also made of steel, so a ferrous particle sensor is used to monitor axles in construction equipment in this study. The general scheme of a ferrous particle sensor is shown in Figure 3; it is composed of two units, where; each unit has an inductance sensing coil and a permanent magnet for collecting ferrous debris at the sensor tip. This sensor is characterized by its ability to discriminate between fine and coarse ferrous debris. Moreover, this sensor can prevent three-body abrasive wear and wear damage by the accumulation of ferrous debris owing to the permanent magnet. This type of sensor is mainly used for detecting ferrous debris in various mechanical systems, such as the gearboxes in wind-turbines, and engines.

In the process of developing wear particle sensors, there are studies on the sensitivity of the sensor [5,30,31,32,33]. However, there are only a few pieces of research that introduce research results using ferrous particle sensors applied to actual lubrication systems. Moreover, there is only a study on the sensitivity of the sensor by the effect of debris position [34], but it is too limited, and there is no study on the location of the sensor to improve sensitivity. Therefore, we applied it to a gear system suitable for condition diagnosis using a ferrous particle sensor and tried to identify its utilization method and problems. In addition, we suggested a method for selecting the location of the sensor to improve its sensitivity. 

We developed a monitoring program based on a ferrous particle sensor and a simple diagnostic algorithm. The monitoring system was installed on the durability test device of the axle of the construction equipment to diagnose the condition. During diagnosis, there was a case where the ferrous particle sensor could not measure abnormalities despite axle failure. Thus, sensor positioning is very important for diagnosing the machine’s condition. Accordingly, a numerical approach for the location selection of the sensor is introduced in this study.

## 2. Experimental Device and Method

Experiments were performed on a bench test device for the axles of construction equipment, as shown in Figure 4. The shaft is rotated by an electric motor that operates at about 514–1370 rpm, and loads are applied to both hubs, which act vertically upward from below. The tests were performed on the axle of an 8-ton grade forklift. The axle had sump-type lubrication, so the ferrous particle sensor was installed at the drain port. Figure 5 shows the measured part used with the ferrous particle sensor. Arduino was used to convert signals from analog to digital format; the signal conversion device was enclosed in a 3-D printed case to allow connection with the signal processing part of the sensor. The specifications of the ferrous particle sensor are shown in Table 1. The lubricant oil is gear oil and Spirax S4 TXM from SHELL; the lubricant properties are shown in Table 2. 

The sensor output is either a voltage or a current signal, and in this experiment, it was set to output a voltage signal. To determine the amount of ferrous wear according to the voltage signal from the sensor, a relational expression between the number of ferrous wear particles and the voltage signal was derived. First, standard specimens for fine and coarse particles were prepared, as shown in Figure 6, because it was not easy to separate the ferrous particles from the sensor measuring part owing to the permanent magnet in the sensor. For the fine particles, 45 μm-sized ferrous particle powder was used, and for the coarse particles, steel balls of diameter 2 mm were used. After injecting the particles into the crystal-resin, the polymer was hardened at room temperature and used as a standard specimen.

Figure 7 shows the linear regression between the weight of ferrous particles and voltage. The root mean squares of the linear fitting in Figure 7a,b were 0.983 and 0.974, respectively. Equations (1) and (2) show the relationships between the weight and voltage for fine and coarse particles, respectively, where; *W* (mg) is the average weight of ferrous particles and V is the voltage.
*W* = 41.62 × V − 43.57(1)
*W* = 118.65 × V − 186.87(2)

The main component of the axle housing is iron, and the drain port where the sensor is installed on the axle is also made of iron. The variation in the measured voltage because of the surrounding iron was investigated using a nut of a size similar to the drain port, as shown in Figure 8. That is, a simple test was performed to determine the degree of interference from the iron component of the mechanical elements around the ferrous particle sensor, and these results are shown in Table 3. The case without any iron-based mechanical element around the sensor was considered as the standard, whose measured voltage was 1.0 V. The measurements error with interference were not large and were within 2%. Therefore, the influence of the drain port on the sensor was considered to be negligible.

A diagnostic program was developed using Java language to easily monitor the axle condition, as shown in Figure 9. When the user clicks the run icon (execution icon on the computer screen), a set-up screen appears, as shown in Figure 9a. The threshold values related to the weight of ferrous particles and gradient with time should be entered in the set-up screen. In addition, after setting the communication port, the set button is clicked to run the main screen, as shown in Figure 9b. 

Of the two graphs on the main screen, the graph on the left shows the measured voltage signals for the fine and coarse particles; here, the solid black and blue lines represent the voltage signals for the fine and coarse particles, respectively. Moreover, the orange dotted and red dash-dot lines indicate the thresholds for the caution and danger conditions, respectively. The graph on the right shows the weights of the fine and coarse particles; the voltage signals were converted to the weights of the particles using Equations (1) and (2). The program provides a warning when the measured values exceed the given thresholds. The warning system indicates the condition through a message or corresponding color, similar to a traffic light. The conditions were largely divided into three stages: normal (green), caution (yellow), and danger (red). For example, when the fine particles in the lubricant suddenly increase, the solid black line changes as shown in Figure 9b. In the figure, when the quantity of fine particles exceeds the threshold of the caution stage, a yellow warning light and warning message are displayed to indicate that the slope is approaching dangerous levels. Additional analyses are facilitated through a function that separates and stores the information in the area where the measured data exceed the threshold. 

## 3. Results and Discussion

The durability test of the axle was performed using the condition monitoring system with the ferrous particle sensor. Constant vertical loads of 2.2 tons were each applied to both hubs, and the shaft was rotated at different speeds from 514 to 1370 rpm according to the gear shift stage. The tests were performed for the forward as well as the reverse rotation. The lubricant used was new oil, and a single ferrous particle sensor was applied to the axle. 

### 3.1. Experimental Results

The durability tests were conducted on two types of newly designed axles during the development process. These newly designed axles are expected to be applied to 8-ton grade forklifts once they pass the durability and field tests. Figure 10 shows the experimental result of the first case; the graph on the left is the result of the overall experiment, and the blue dotted box indicates the abnormal condition. The graph on the right is an enlarged view of the results of the abnormal condition. The test was performed continuously for about 47 h until the axle gears broke, and severe wear or breakage occurred 2.5 min after onset. There was no increase in the amount of ferrous-wear particles before the severe breakage. At the point where the wear amount increased suddenly, the rate of change of the voltage with time was close to 1, i.e., the voltage increased almost vertically. Further durability tests were not possible owing to the sudden destruction of the gears in the axle. The damage to the gears in the axle and ferrous-wear particles attached to the sensor are shown in Figure 11. In Figure 11a, the gears in the axle are observed to be destroyed to such an extent that their shapes could not be confirmed. Figure 11b shows a large quantity of ferrous-wear particles gathered at the measuring part of the sensor. 

The durability test of the second type of newly designed axle was conducted similarly. Figure 12 shows the experimental result of the second case, whose load and rotational speed conditions are the same as those in the first case. 

### 3.2. Problem of Condition Monitoring with Ferrous Particle Sensor in the Axle

In the second case, the signal was measured to be normal, as shown in Figure 12. However, the gears in the axle were heavily damaged, similar to that in Figure 11a. The reason for this difficulty in diagnosing the axle condition using the ferrous particle sensor is that the wear of the gear could not be measured at the location where the sensor was installed. That is, it was impossible to measure wear because the sensor position was far from the location of wear of the gears. Figure 13 shows the relationship between flow and sensor position for two representative cases to explain the importance of sensor positioning. In the sump-tank-type lubrication method, such as that used in a gearbox, the direction of the main flow varies according to the shape of the gear. The axles of construction equipment are also lubricated by the sump-type method. As shown in Figure 13a, when the main flow by the gear is away from the sensor, it is difficult to measure wear particles. However, when the main flow is around the sensor, as shown in Figure 13b, it is easy to detect the wear particles of the gear. Therefore, in sump-type lubrication, a method of effectively diagnosing the condition through the lubricant sensor is needed. The authors believe that there are two main directions for improving this. The first is to improve the sensitivity of the sensor, and the other is to create a flow artificially around the sensor to facilitate measurement. In the case of the ferrous particle sensor, the quantity of ferrous particles is measured by collecting these particles using a permanent magnet placed within the sensor. Therefore, it is believed that simply increasing the magnetic field strength of the permanent magnet can help collect and measure ferrous-wear particles well; however, it may still be difficult to measure the amount of ferrous-wear particles exactly because of interactions with the ferrous components of the mechanical elements in the immediate surroundings. One may also consider a scheme using simple structures so that the main flow occurs around the sensor. However, it is a good approach to locate the sensor in a position where easy detection is possible once the main flow has been determined from the rotation of the gears. In this study, we introduced a numerical approach of sensor positioning from among all the available methods to improve the measurements by the ferrous particle sensor in sump-type lubrication. The research results on the exact numerical method used and its optimization will be introduced in a subsequent paper.

### 3.3. Numerical Approach of Sensor Positioning

From the perspective of the aforementioned solution without considering the flow caused by the movements of various gears in the actual axle, a simple method is introduced here. This is not an exact solution to the problem, but rather an approach to solving it. The commercial multi-physics software, COMSOL 6.0 was used for the calculations. The electromagnetic field interface model, particle tracing module, and Navier–Stokes equations are used in the numerical analysis. The electromagnetic field interface model is included in the COMSOL AC/DC module, and this model is used to calculate the magnetic flux of the ferrous particle sensor. The particle tracing module is a numerical method of computing the paths of individual particles by solving their equations of motion over time; this module solves a number of discrete trajectories. Using this module, it is possible to assess whether the ferrous particle sensor captures the trajectories of the ferrous particles in the flow well. Figure 14 shows the numerical model of the ferrous particle sensor; it is difficult to obtain accurate design specifications, such as the material of the Gill sensor used in the experiments. The major design parameters were roughly investigated by the sensor disassembly, and the roughly measured design parameters are reflected in the analysis. Moreover, the material of the magnetic core is low-carbon steel M-50, and the B-H curve uses the data given in the analysis program. The sensor was analyzed using a two-dimensional axisymmetric model. Figure 15 shows the numerical model for sensor positioning; all faces except the inlet and outlet are set as stationary walls. Various elements such as tetrahedron, pyramid, prism, hexahedron, and quad were used in the numerical analysis. The total number of elements used is 476,338, and a dense mesh was applied to the measurement area of the sensor; this analysis was conducted using a planar symmetric model. The size of the flow channel is 70 mm × 70 mm × 150 mm, and the flow is laminar. The working conditions of the numerical calculations are shown in Table 4. The particles used in the analysis are spherical and the material is iron with a density of 8030 kg/m^3^. The properties of the lubricant used in the analysis are shown in Table 2. In Figure 15a, S_position indicates the distance from the bottom to the measuring part of the sensor; as the value of S_position increases, the distance from the bottom to the sensor increases, along with an increase in the area of the sensor exposed to the lubricant. 

Figure 16 shows the magnetic flux density when the S_position values are 10 mm and 20 mm; the unit of magnetic flux density is Tesla (T). These results are shown along the xz plane passing through the center of the sensor in the x-axis direction. It can be confirmed that the shape of the magnetic force lines in the flow field changes according to the position of the sensor. Since the shape of the magnetic force lines in the flow varies depending on the location of the sensor, the capture of ferrous particles by the permanent magnet of the sensor will also change. Therefore, it is necessary to check the trajectories of the particles. Figure 17 presents the variation of the particle trajectories with time when the S_position values are 10 mm and 20 mm. A total of 100 ferrous particles flow from the inlet side to the outlet; the shape of the particles is assumed to be spherical, with a diameter of 10 μm, as shown in Table 4. When S_position is 10 mm, there are four ferrous particles attached to the sensor and when S_position is 20 mm, there are three particles attached to the sensor. Obviously, the difference in the number of particles collected owing to the change of S_position is as small as 1; however, it is meaningful that the effects of collecting ferrous particles are different even with a change in only the area of the sensor exposed to the flow. In this study, we do not consider the optimal sensor location but introduce only an analytical approach for selecting the sensor location. As noted in Section 3.2, to optimize the position of the ferrous particle sensor on the axle, the flow analysis due to the motions of the gears as well as a flow design for generating the main flow around the sensor are required. Additional numerical studies on these aspects are still underway. 

## 4. Conclusions

In this study, a condition monitoring system comprising a ferrous particle sensor was developed to monitor the condition of axles in construction equipment. The monitoring system was applied to a bench test device for durability evaluations. The sensor is composed of an inductance sensing coil and a permanent magnet for collecting ferrous wear debris. Moreover, the sensor can prevent three-body abrasive wear because of the permanent magnet. Ferrous wear particles are generated owing to metal-to-metal contact in the axle, so the sensor is suitable for diagnosing abnormalities. In the industrial field, the condition was diagnosed using the functions provided by the manufacturer of the ferrous particle sensor, and there are no studies on the effect of sensor position. Most of the research focuses on improving the internal design factors of the sensor to improve the sensitivity. So, in industrial applications, the influence of the sensitivity by the position of the sensor has not been considered, and there are frequent cases in which abnormalities are not observed despite the failure of the mechanical system. In this study, a diagnostic program was additionally developed to effectively monitor the axle condition. The program includes graphs showing the numbers of ferrous particles, variation (slope) of wear amount, and original voltages for fine and coarse particles. Moreover, warning systems are implemented to display warning messages, with colors similar to those of traffic lights. In addition, to facilitate analysis, a function is provided to separate and store the abnormal data measurements that exceed the threshold values. The durability tests were performed for two cases of axle types. In the first case, sudden abnormal wear occurred, resulting in severe damage to the gears in the axle, for which the diagnostic program measured a rapid increase in the amount of wear in about 2.5 min. In the second case, the diagnostic program did not measure such changes even though the axle was greatly damaged by the abnormal wear of the gears; the reason for this is that the sensor position was unsuitable for measuring the wear of the gears in the axle. Therefore, the appropriate positioning of the ferrous particle sensor is necessary for the effective diagnosis of the axle. This study also introduces a numerical analysis method for the optimal location selection of the sensor. The electromagnetic field interface model, the particle tracing module, and Navier–Stokes equations were used in the numerical analysis, and the multi-physics problem was evaluated using commercial software. The two cases were analyzed, and the results showed the effects of collecting ferrous particles and the magnetic flux density of the sensor based on the sensor location. This is meaningful in that the proposed numerical approach for ferrous particle sensor location selection allows effective monitoring of the lubrication system. 

To accurately select the sensor location in the axle, it is necessary to analyze the flow within the axle because the main flow affects the sensor measurements. Moreover, it is necessary to design the flow such that the main flow occurs around the sensor. Hence, additional studies must be conducted to improve the insufficiencies of this work by complementing the analysis method proposed herein.

## Figures and Tables

**Figure 1 materials-16-01426-f001:**
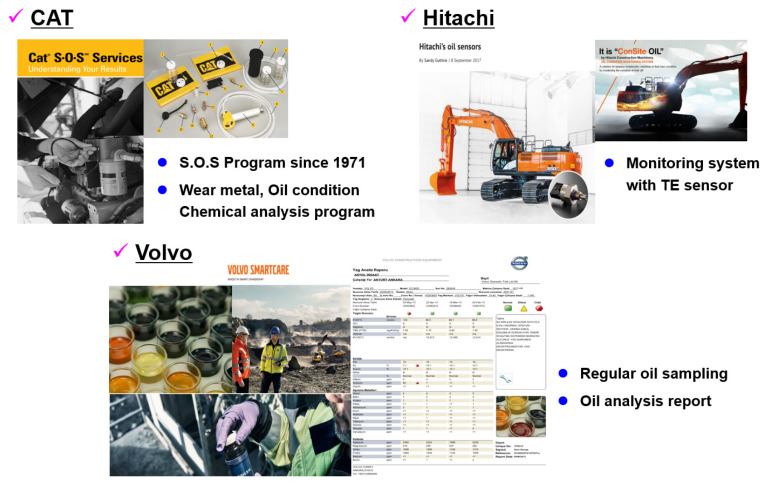
Oil analysis service of construction equipment company.

**Figure 2 materials-16-01426-f002:**
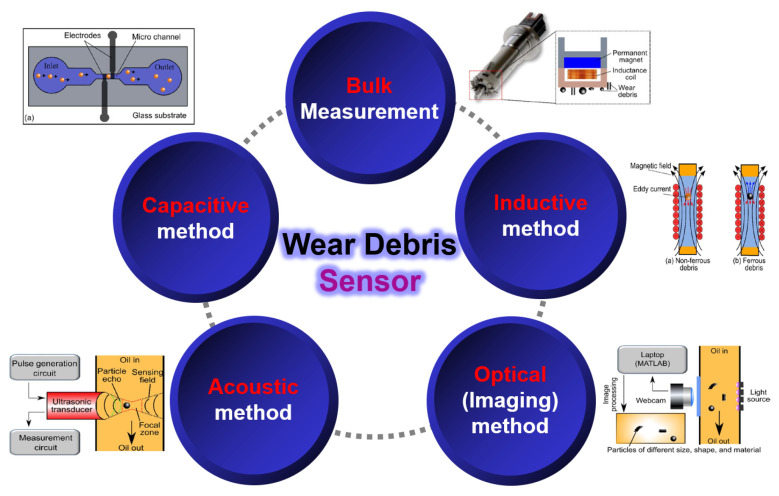
Various types of wear debris sensors.

**Figure 3 materials-16-01426-f003:**
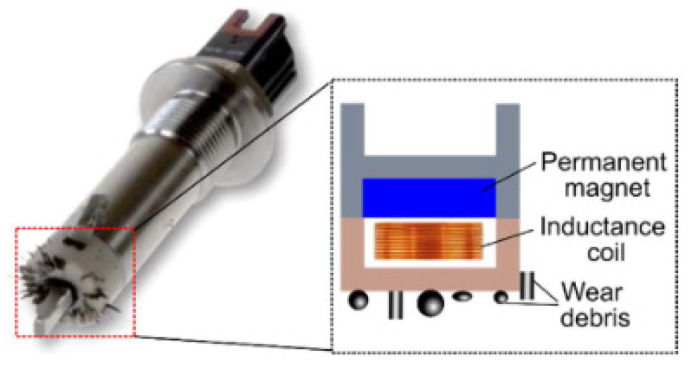
Schematic of ferrous particle sensor developed by Gill sensors [21].

**Figure 4 materials-16-01426-f004:**
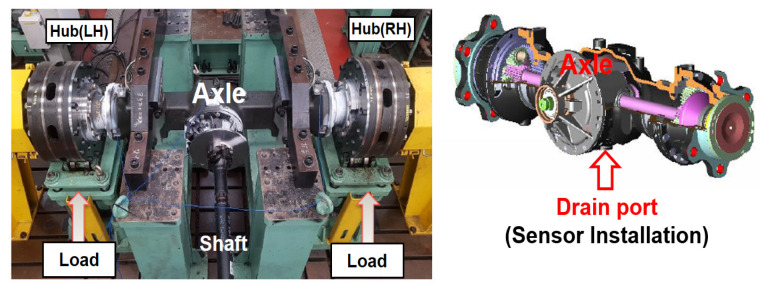
Bench test device of the axle.

**Figure 5 materials-16-01426-f005:**
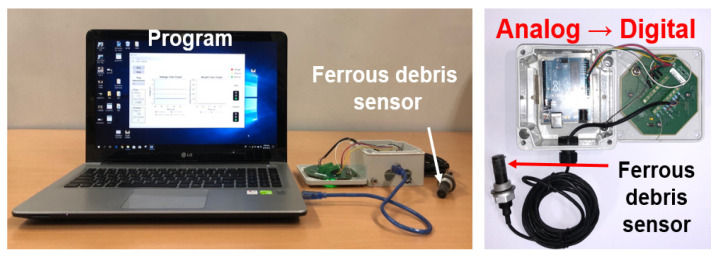
Measured part connected to the ferrous particle sensor.

**Figure 6 materials-16-01426-f006:**
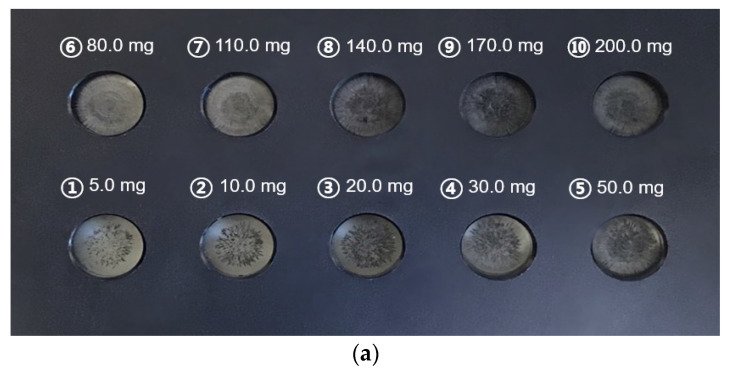
Standard specimens of the ferrous particles: (**a**) Fine particles; (**b**) Coarse particles.

**Figure 7 materials-16-01426-f007:**
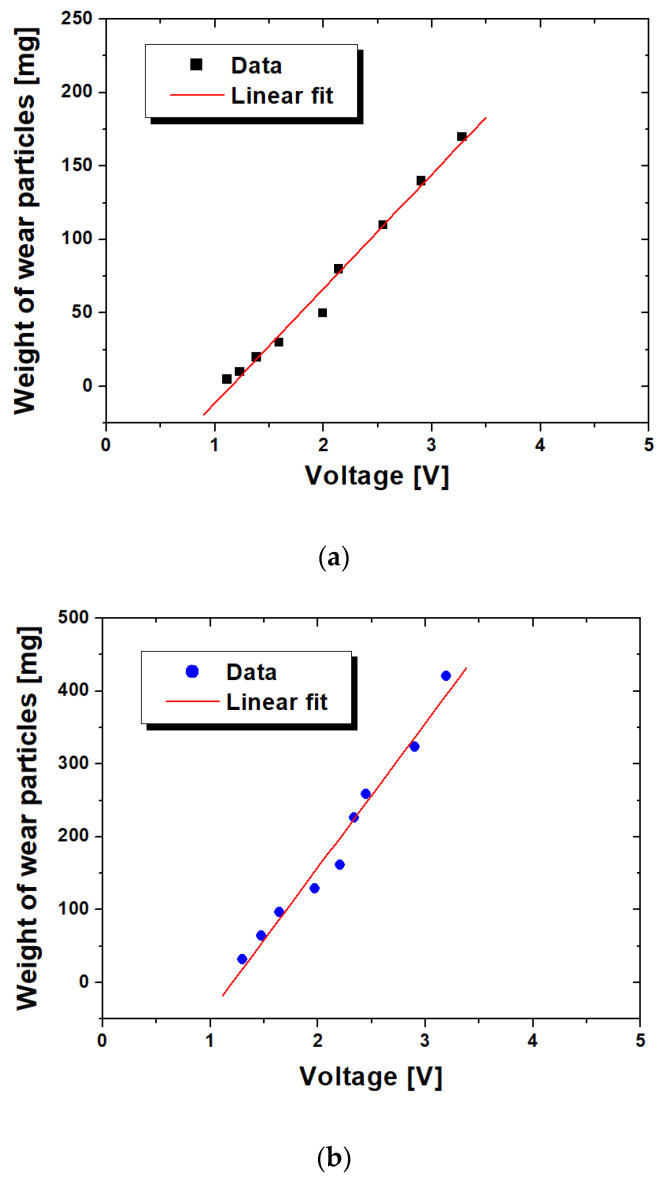
Linear regressions of wear amount against voltage: (**a**) Fine particles; (**b**) Coarse particles.

**Figure 8 materials-16-01426-f008:**
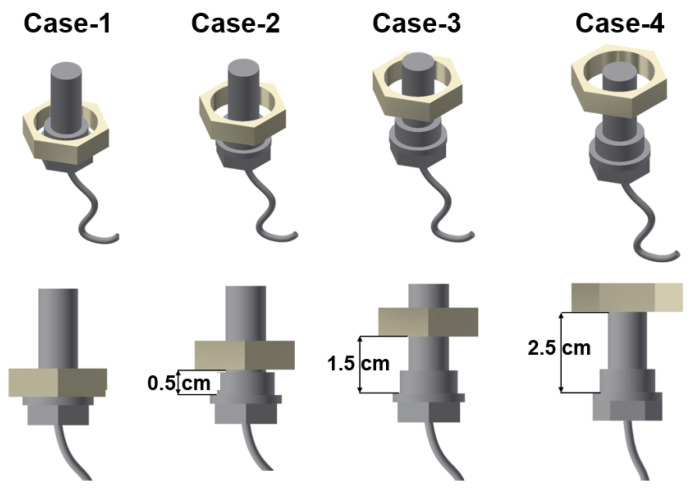
Location of the sensor with respect to the nut.

**Figure 9 materials-16-01426-f009:**
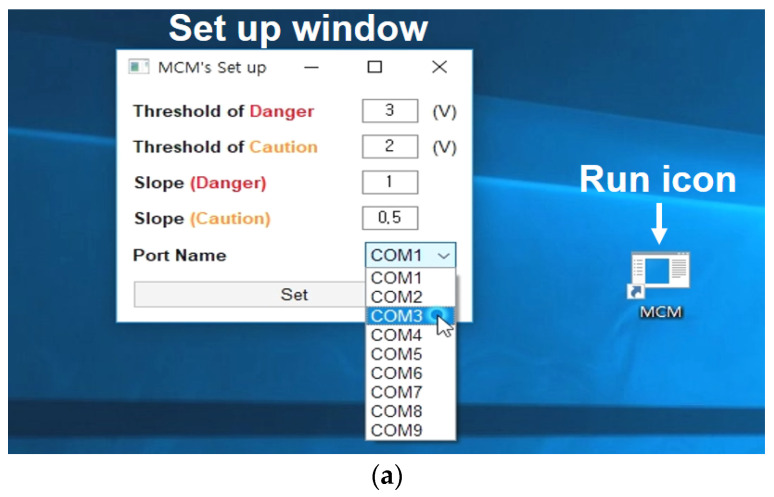
Diagnostic program: (**a**) Set-up window; (**b**) Main window.

**Figure 10 materials-16-01426-f010:**
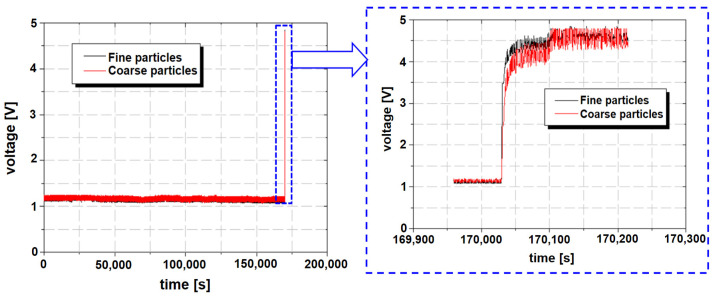
Measured voltage with time in the first case.

**Figure 11 materials-16-01426-f011:**
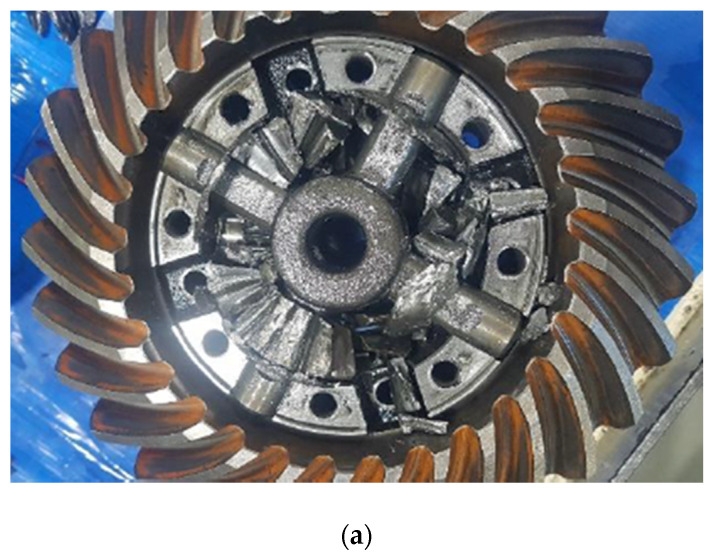
Broken axle and ferrous wear particles: (**a**) Gear breakage at the axle; (**b**) Ferrous wear particles attached to the sensor.

**Figure 12 materials-16-01426-f012:**
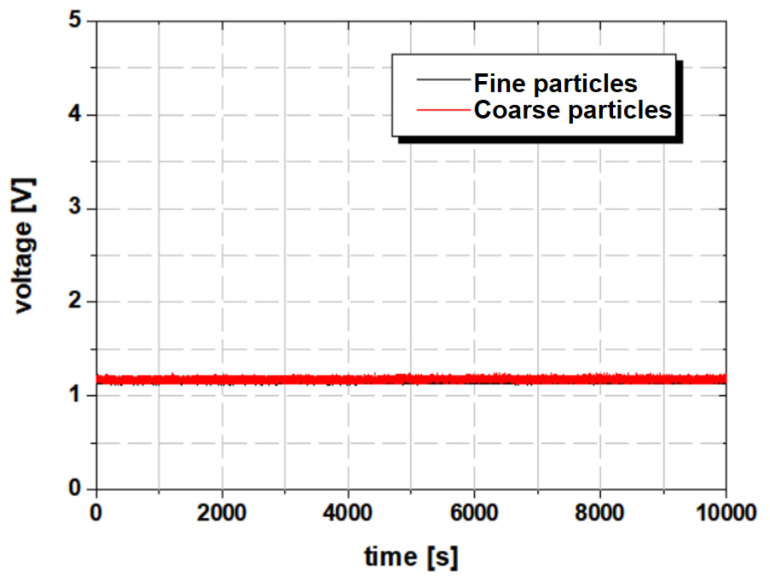
Measured voltage with time in the second case.

**Figure 13 materials-16-01426-f013:**
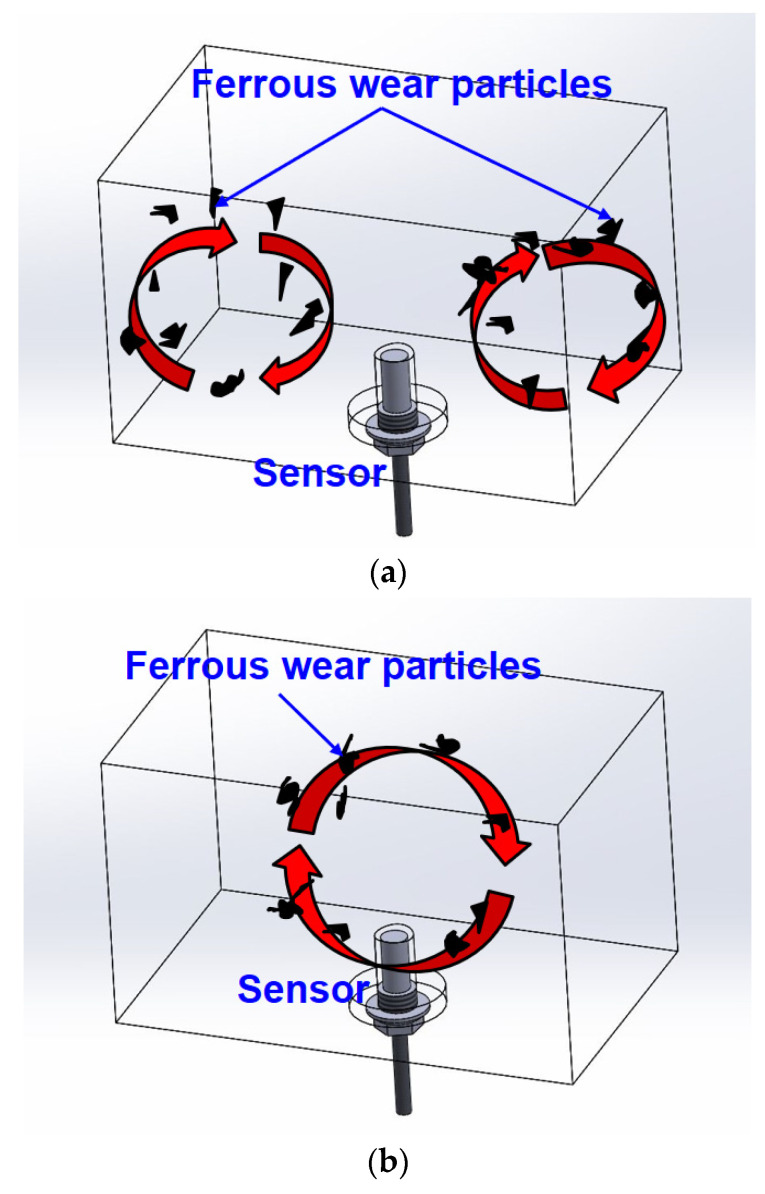
Relationship between flow and sensor position: (**a**) Case where it is difficult to detect wear particles; (**b**) Case where it is easy to detect wear particles.

**Figure 14 materials-16-01426-f014:**
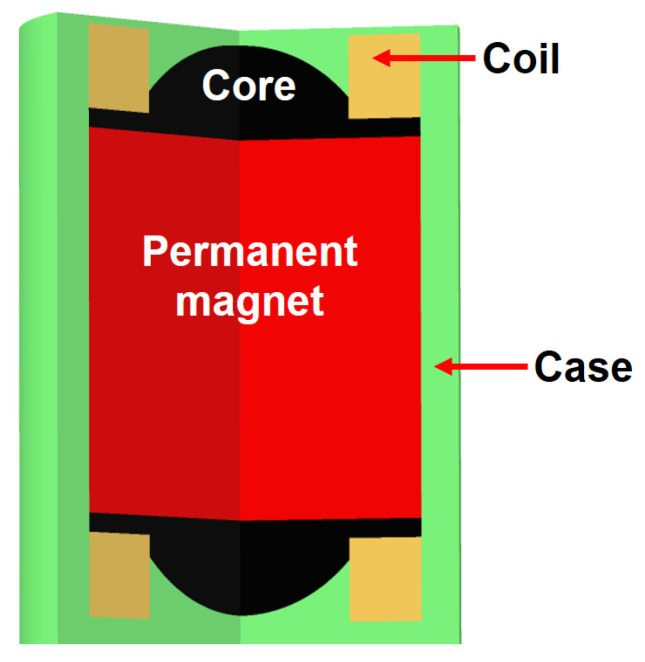
Numerical model of the ferrous particle sensor.

**Figure 15 materials-16-01426-f015:**
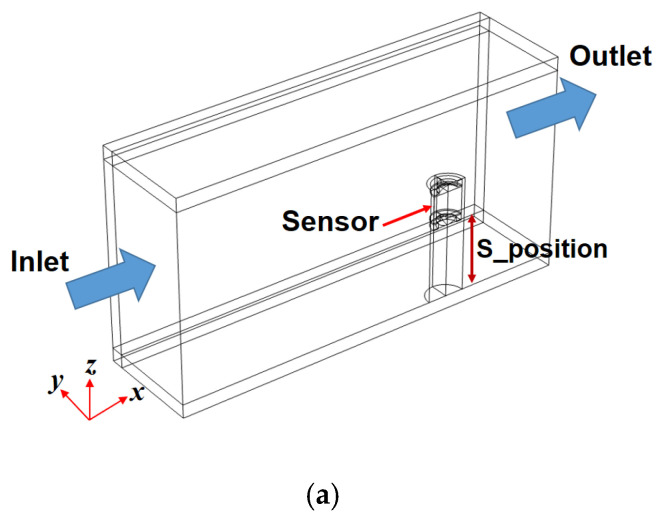
Numerical model used for sensor positioning: (**a**) Control volume; (**b**) Mesh.

**Figure 16 materials-16-01426-f016:**
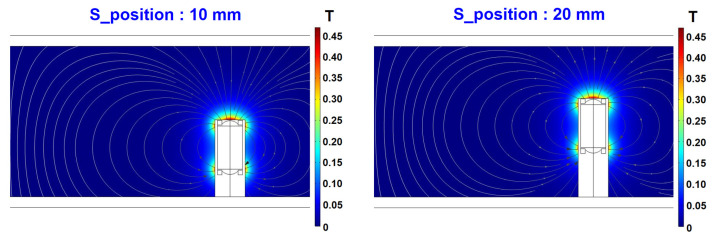
Magnetic flux density in the xz plane with variation of S_position.

**Figure 17 materials-16-01426-f017:**
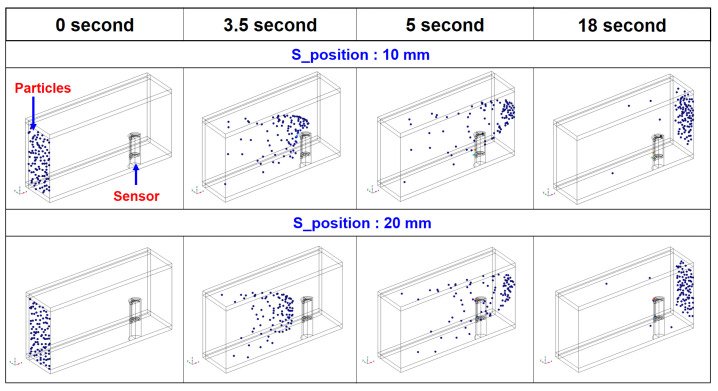
Particle trajectories with time.

**Table 1 materials-16-01426-t001:** Specifications of the ferrous particle sensor.

Items	Function	Items	Function
Working temperature	–40 to 180 °C	Weight	25 g
Working pressure	Max. 10 bar	Supply voltage	4.5 V–32 V (DC)
Fluid compatibility	Petroleum oils, General automotive fluids	Channel 1	Fine particles (2.25 V–4.25 V)
Sampling rate	10 Hz	Channel 2	Coarse particles (0.5 V–4.25 V)

**Table 2 materials-16-01426-t002:** Properties of the test lubricant.

Properties	Method	Value
SAE Viscosity Grade	SAE J 300	10 W-30
Kinematic viscosity @ 40 °C	ISO 3104	60 mm^2^/s
Kinematic viscosity @ 100 °C	ISO 3104	9.4 mm^2^/s
Viscosity Index	ISO 2909	138
Density @ 15 °C	ISO 12185	882 kg/m^3^
Flash Point (COC)	ISO 2592	220 °C
Pour Point	ISO 3016	−42 °C

**Table 3 materials-16-01426-t003:** Variation in measured voltage according to the relative locations of the sensor and nut.

Case	Standard	Case-1	Case-2	Case-3	Case-4
Voltage	1.0 V	1.02 V	1.02 V	1.02 V	1.0 V

**Table 4 materials-16-01426-t004:** Working conditions for the numerical calculations.

Items	Condition	Items	Condition
Inlet	Inlet velocity 0.02 m/s	Outlet	Outflow
Number of particles	100	Particle diameter	10 μm
Shape of particle	Sphere	Material of particle	Steel
Particle relative permeability	1000	Particle density	8030 kg/m^3^

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
