# Peer review of "Assessment of Condition Diagnosis System for Axles with Ferrous Particle Sensor"

_materials, 2023, doi:10.3390/ma16041426_

Round 1

Reviewer 1 Report

In this paper, the authors developed a condition monitoring system using a ferrous particle sensor to monitor the axels condition in heavy machinery. In addition, they created a program demonstrating the amounts of ferrous particles and associated voltages for both small size and large size particles. I do not think the current version of the manuscript should be published in Materials since the novelty, significance of content, scientific soundness and overall merit is low. Following are some specific comments regarding the manuscript:

• What is the main question addressed by the research? The main contribution of this manuscript is to monitor the amount of generated ferrous particles (debris) in axels due to metal-to-metal contact using a commercial simple sensor.

• Do you consider the topic original or relevant in the field? Does it address a specific gap in the field?

No, this topic is not original. Structural health monitoring of moving components, particularly metallic, is critical however this manuscript does not contribute to the literature since the authors used a commercial sensor and only developed a program to monitor the amount of generated ferrous particles.

• What does it add to the subject area compared with other published material?

I believe this manuscript has severely limited addition to the subject. The authors used a commercial sensor and developed a program to measure the amount of generated particles due to metal-to-metal contact in axels. The presented work in the manuscript is not novel and overall merit is very low. In addition, the authors attempted to study the impact of sensor location on sensitivity and efficiency of the monitoring condition however only studied two scenarios including easy detection of the wear particles and difficult to detect debris. Although this shows the importance of sensor location, however, does not address whether this is the best design and can be adopted to other axels too.  

Author Response

Dear Editor and Reviewer

We appreciate your comments.

We prepared sincere responses to your comments as follows.

In this paper, the authors developed a condition monitoring system using a ferrous particle sensor to monitor the axels condition in heavy machinery. In addition, they created a program demonstrating the amounts of ferrous particles and associated voltages for both small size and large size particles. I do not think the current version of the manuscript should be published in Materials since the novelty, significance of content, scientific soundness and overall merit is low. Following are some specific comments regarding the manuscript:

Answer: The ferrous particle sensor, which is widely used in the industry, diagnoses the condition according to the voltage or current output provides by the manufacturer. In addition, there are frequent cases where abnormality is not detected despite a failure. Therefore, this study identifies the working principle of the ferrous particle sensor, develops an additional diagnostic program, and presents a method for more effective condition diagnosis. Furthermore, a method for selecting the location of the sensor is presented. Of course, the optimal method for selecting the location of the sensor for all cases is not presented, but additional research results are being prepared. Numerical analysis considering the flow by the movement of the gears are being performed to select proper location of the sensor, and additional papers will be presented. It is regrettable that all the results mentioned above could not be written in this paper.

However, this study first presented the problem when the ferrous particle sensor is applied to condition diagnosis of gear systems. In addition, in order to accurately identify the problem, after identifying the previous studies on improving sensitivity as a method of changing the internal design of the sensor, we find that there are no studies about selecting position of the sensor to improve sensitivity. Unlike previous studies, a method using multi-physics was presented for the first time. Of course, as mentioned before, the fact that the optimal results were not presented will be presented in another paper as more specific research results.

We revised manuscript in consideration of the reviewer’s comments. In addition to explanations for the deficiencies of previous studies, a detailed explanation was added. We would appreciate it if you could review the revised manuscript once more.   

Point-1:  What is the main question addressed by the research? The main contribution of this manuscript is to monitor the amount of generated ferrous particles (debris) in axels due to metal-to-metal contact using a commercial simple sensor.

 Answer: We used ferrous particle sensor to diagnose condition of the axle in construction equipment.

Point-2:  Do you consider the topic original or relevant in the field? Does it address a specific gap in the field?

No, this topic is not original. Structural health monitoring of moving components, particularly metallic, is critical however this manuscript does not contribute to the literature since the authors used a commercial sensor and only developed a program to monitor the amount of generated ferrous particles.

 Answer: Until now, the degree of damage was known after the experiment, but using this sensor, a system that can prevent large damage by using the overall amount of wear or by checking the increasing slope of wear was developed and applied. A method for effectively diagnosing the condition of a gear system using ferrous particle sensor was also presented. In addition, the sensitivity of measurement is greatly affected by the position of the sensor Therefore, it is considered to have originality in that it was the first to propose a method to improve it. So, we added explanations as follows.

Title

“Assessment of Condition Diagnosis System for Axles with Ferrous Particle Sensor”

Abstract

“In the two tests, the first case detected a failure, but in the other case, the sensor did not detect even though a failure occurred.”

Introduction

“In axles, most of them show adhesive wear, abrasive wear and fatigue wear.”

“Sensors for measuring wear particles are being developed in various ways, but most of them have been developed based on inductance and capacitance. Besides oil condition diagnosis based on wear particle sensors, oil condition diagnosis using a complicated sensor that measures various properties at once with one sensor is also applied [3, 28].”

“Wear is a phenomenon closely related to machine condition, maintenance, reliability. Therefore, it is important to determine the amount or type of wear [29]. It is also important to identify the particles generated by wear in lubricating systems.”

“In the process of developing wear particle sensors, there are studies on the sensitivity of the sensor [5, 30-33]. However, there are few researches that introduce research results using ferrous particle sensors applied to actual lubrication systems. Moreover, there is a study on the sensitivity of the sensor by the effect of debris position [34], but it is too limited research and there is no study on the location of the sensor to improve sensitivity. Therefore, we applied it to a gear system suitable for condition diagnosis using a ferrous particle sensor and tried to identify its utilization method and problems. In addition, we suggested a method for selecting the location of the sensor to improve the sensitivity of it.”

Experimental Device and Method

“The root mean squares of the linear fitting in Figure 7 (a) and (b) were 0.983 and 0. 974, respectively.”

Results and Discussion

“Various elements such as tetrahedron, pyramid, prism, hexahedron and quad were used in the numerical analysis. The total number of elements used is 476,338, and a dense mesh was applied to the measurement area of the sensor.”

Conclusions

“In the industrial field, the condition was diagnosed using the functions provided by the manufacturer of the ferrous particle sensor, and there are no studies on the effect of sensor position. Most of the researches focus on improving the internal design factors of the sensor to improve the sensitivity. So, in the industrial applications, the influence of the sensitivity by position of the sensor has not been considered, and there are frequent cases in which abnormalities are not observed despite the failure of the mechanical system.

References

We added more references.

  1. Li, W.; Bai, C.; Wang, C.; Zhang, H.; Ilerioluwa, L.; Wang, X.; Yu, S.; Li, G. Design and research of inductive oil pollutant detection sensor based on high gradient magnetic field structure. Micromachines 2021, 12, 638.
  2. Feng, S.; Yang, L.; Qiu, G.; Luo, J.; Li, R.; Mao, J. An inductive debris sensor based on high-gradient magnetic field. IEEE Sens. J. 2019, 19, 2879-2886.
  3. Wu, S.; Liu, Z.; Yu, K.; Fan, Z.; Yuan, Z.; Sui, Z.; Yin, Y.; Pan, X. A novel multichannel inductive wear debris sensor based on time division multiplexing. IEEE Sens. J. 2021, 21, 11131-11139.
  4. Hong, W.; Li, T.; Wang, S.; Zhou, Z. A general framework for aliasing corrections of inductive oil debris detection based on artificial neural networks. IEEE Sens. J. 2020, 20, 10724-10732.
  5. Muthuvel, P.; George, B.; Ramadass, G.A. A highly sensitive in-line oil wear debris sensor based on passive wireless LC sensing. IEEE Sens. J. 2021, 21, 6888-6896.
  6. Muthuvel, P.; George, B.; Ramadass, G.A. Magnetic-capacitive wear debris sensor plug for condition monitoring of hydraulic systems. IEEE Sens. J. 2018, 18, 9120-9127.

  1. Jeon, H. G.; Kim, J.K.; Na, S.J.; Kim, M. S.; Hong, S. H. Application of condition monitoring for hydraulic oil using tuning fork sensor: a case on hydraulic system of earth moving machinery. Materials 2022, 15, 7657.
  2. Fasihi, P., Kendall, O.; Abrahams, R.; Mutton, P.; Qiu, C.; Schlafer, T.; Yan, W. Tribological properties of laser cladded alloys for repair of rail components. Materials 2022, 15, 7466.
  3. Ren. Y.; Li, W.; Zhao, G.; Feng, Z.; Inductive debris sensor using one energizing coil with multiple sensing coils for sensitivity improvement and high throughput. Tribol. Int. 2018, 128, 96-103.
  4. Xiao, H.; Wang, X.; Li, H.; Luo, J.; Fong, S. An Inductive debris sensor for large-diameter lubricating oil circuit based on a high-gradient magnetic field. Appl. Sci. 2019, 10, 1546.
  5. Ma, L.; Zhang, H.; Qiao, W.; Han, X.; Zeng, L.; Shi, H. Oil metal debris detection sensor using ferrite core and flat channel for sensitivity improvement and high throughput. IEEE Sens. J. 2020, 20, 7303-7309.
  6. Zeng, L.; Yu, Z.; Zhang, H.; Zhang, X.; Chen, H. A high sensitive multi-parameter micro sensor for detection of multi-contamination in hydraulic oil. Sens. Actuators A Phys. 2018, 282, 197-205.
  7. Ma, L.; Zhang, H.; Zheng, W.; Shi, H.; Wang, C.; Xie, Y. Investigation on the effect of debris position on the sensitivity of the inductive debris sensor. IEEE Sens. J. DOI: 10.1109/JSEN.2022.3155256.

Point-3: What does it add to the subject area compared with other published material?

I believe this manuscript has severely limited addition to the subject. The authors used a commercial sensor and developed a program to measure the amount of generated particles due to metal-to-metal contact in axels. The presented work in the manuscript is not novel and overall merit is very low. In addition, the authors attempted to study the impact of sensor location on sensitivity and efficiency of the monitoring condition however only studied two scenarios including easy detection of the wear particles and difficult to detect debris. Although this shows the importance of sensor location, however, does not address whether this is the best design and can be adopted to other axels too.  

Answer: In actual gear systems, it is difficult to diagnose the condition due to changes in the physical properties of the lubricating oil such as viscosity and acid number, and even in the condition diagnosis using the ferrous particle sensor, it is often difficult to diagnose abnormalities despite the failure due to the sensor's sensitivity limit. There are many studies on improving the design factor inside the sensor to improve the sensitivity of the sensor, but it is judged that the positioning of the sensor is more effective in a large lubrication system such as an actual gear system. So, in this respect, a method for selecting the location of a sensor using multi-physics was proposed. Moreover, there are still many research results on the development of wear particle sensors, but it is judged that there are few successful cases of actual application in the industry. In this respect, it is a study using commercially used sensors, but I think it is meaningful in that it mentions real problems and suggests a methodology to improve them.

We appreciate your comments during the first paper review process.

It is thought that the completeness of our paper has been improved more through the correction of each comment.

Thank you once again for reviewing our paper.

Sincerely yours,

Sung-Ho Hong

Reviewer 2 Report

Please improve the title.

Suggested title: Assessment of Condition Diagnostic system for dual- axles with ferrous particle sensor

Mesh information: No. of element, type of element etc. should be given

Please add more relevant literatures. No of literatures cited is very less.

1. What is the main question addressed by the research? Res# Ferrous Particle Sensor used to address condition diagnostics
2. Do you consider the topic original or relevant in the field? Does it
address a specific gap in the field? Res# Yes
3. What does it add to the subject area compared with other published
material? Res#a multi-physics-based analysis method is suggested for positioning the ferrous particle sensor.
4. What specific improvements should the authors consider regarding the
methodology? What further controls should be considered? Res# 1. Title changed to "Assessment of Condition Diagnostic system for dual-axles with ferrous particle sensor", 2. Mesh information: No. of elements, type of element, etc. should be given, 3. Please add more relevant literatures. No of literature cited are very less.
5. Are the conclusions consistent with the evidence and arguments presented
and do they address the main question posed? Res #Author should improve by stating the limitations, and application of the current method.
6. Are the references appropriate? Resp# More references should be added
7. Please include any additional comments on the tables and figures. Res# Please improve the resolution of the figures.

Author Response

Dear Editor and Reviewer

We appreciate your comments.

We prepared sincere responses to your comments as follows.

Point-1: Please improve the title.

Suggested title: Assessment of Condition Diagnostic system for dual- axles with ferrous particle sensor

Answer: We improved the title as follows.

“Assessment of Condition Diagnosis System for Axles with Ferrous Particle Sensor”

Point-2: Mesh information: No. of element, type of element etc. should be given

Answer: We added information about number and type of elements as follows.

“Various elements such as tetrahedron, pyramid, prism, hexahedron and quad were used in the numerical analysis. The total number of elements used is 476,338, and a dense mesh was applied to the measurement area of the sensor;”

Point-3: Please add more relevant literatures. No of literatures cited is very less.

Answer: We added relevant literatures as follows.

  1. Li, W.; Bai, C.; Wang, C.; Zhang, H.; Ilerioluwa, L.; Wang, X.; Yu, S.; Li, G. Design and research of inductive oil pollutant detection sensor based on high gradient magnetic field structure. Micromachines 2021, 12, 638.
  2. Feng, S.; Yang, L.; Qiu, G.; Luo, J.; Li, R.; Mao, J. An inductive debris sensor based on high-gradient magnetic field. IEEE Sens. J. 2019, 19, 2879-2886.
  3. Wu, S.; Liu, Z.; Yu, K.; Fan, Z.; Yuan, Z.; Sui, Z.; Yin, Y.; Pan, X. A novel multichannel inductive wear debris sensor based on time division multiplexing. IEEE Sens. J. 2021, 21, 11131-11139.
  4. Hong, W.; Li, T.; Wang, S.; Zhou, Z. A general framework for aliasing corrections of inductive oil debris detection based on artificial neural networks. IEEE Sens. J. 2020, 20, 10724-10732.
  5. Muthuvel, P.; George, B.; Ramadass, G.A. A highly sensitive in-line oil wear debris sensor based on passive wireless LC sensing. IEEE Sens. J. 2021, 21, 6888-6896.
  6. Muthuvel, P.; George, B.; Ramadass, G.A. Magnetic-capacitive wear debris sensor plug for condition monitoring of hydraulic systems. IEEE Sens. J. 2018, 18, 9120-9127.

  1. Jeon, H. G.; Kim, J.K.; Na, S.J.; Kim, M. S.; Hong, S. H. Application of condition monitoring for hydraulic oil using tuning fork sensor: a case on hydraulic system of earth moving machinery. Materials 2022, 15, 7657.
  2. Fasihi, P., Kendall, O.; Abrahams, R.; Mutton, P.; Qiu, C.; Schlafer, T.; Yan, W. Tribological properties of laser cladded alloys for repair of rail components. Materials 2022, 15, 7466.
  3. Ren. Y.; Li, W.; Zhao, G.; Feng, Z.; Inductive debris sensor using one energizing coil with multiple sensing coils for sensitivity improvement and high throughput. Tribol. Int. 2018, 128, 96-103.
  4. Xiao, H.; Wang, X.; Li, H.; Luo, J.; Fong, S. An Inductive debris sensor for large-diameter lubricating oil circuit based on a high-gradient magnetic field. Appl. Sci. 2019, 10, 1546.
  5. Ma, L.; Zhang, H.; Qiao, W.; Han, X.; Zeng, L.; Shi, H. Oil metal debris detection sensor using ferrite core and flat channel for sensitivity improvement and high throughput. IEEE Sens. J. 2020, 20, 7303-7309.
  6. Zeng, L.; Yu, Z.; Zhang, H.; Zhang, X.; Chen, H. A high sensitive multi-parameter micro sensor for detection of multi-contamination in hydraulic oil. Sens. Actuators A Phys. 2018, 282, 197-205.
  7. Ma, L.; Zhang, H.; Zheng, W.; Shi, H.; Wang, C.; Xie, Y. Investigation on the effect of debris position on the sensitivity of the inductive debris sensor. IEEE Sens. J. DOI: 10.1109/JSEN.2022.3155256.

Point-4: 1. What is the main question addressed by the research? Res# Ferrous Particle Sensor used to address condition diagnostics

Answer: We used ferrous particle sensor to diagnose condition of the axle in construction equipment.

Point-5: 2. Do you consider the topic original or relevant in the field? Does it
address a specific gap in the field? Res# Yes

Answer: We added more information about similar studies and originality about our research.

Point-6: 3. What does it add to the subject area compared with other published
material? Res#a multi-physics-based analysis method is suggested for positioning the ferrous particle sensor.

Answer: We suggested new method to select the position of ferrous particle sensor in gear system. In this method, we used multi-physics-based analysis method.

Point-7: 4. What specific improvements should the authors consider regarding the
methodology? What further controls should be considered? Res# 1. Title changed to "Assessment of Condition Diagnostic system for dual-axles with ferrous particle sensor", 2. Mesh information: No. of elements, type of element, etc. should be given, 3. Please add more relevant literatures. No of literature cited are very less.

Answer: We revised title and added information about element and references.

Point-8: 5. Are the conclusions consistent with the evidence and arguments presented
and do they address the main question posed? Res #Author should improve by stating the limitations, and application of the current method.

Answer: We added information about limitations and application of the current method in conclusion as follows.

“In the industrial field, the condition was diagnosed using the functions provided by the manufacturer of the ferrous particle sensor, and there are no studies on the effect of sensor position. Most of the researches focus on improving the internal design factors of the sensor to improve the sensitivity. So, in the industrial applications, the influence of the sensitivity by position of the sensor has not been considered, and there are frequent cases in which abnormalities are not observed despite the failure of the mechanical system.

Point-9: 6. Are the references appropriate? Resp# More references should be added

Answer: We added appropriate references.

Point-10: 7. Please include any additional comments on the tables and figures. Res# Please improve the resolution of the figures.

Answer: We improved the resolution of several figures .

We appreciate your comments during the first paper review process.

It is thought that the completeness of our paper has been improved more through the correction of each comment.

Thank you once again for reviewing our paper.

Sincerely yours,

Sung-Ho Hong

Reviewer 3 Report

Journal: Materials (ISSN 1996-1944)

Manuscript ID: materials-2113709

Review Report 1#

The authors presented an article on “Applications and Problems of Condition Diagnosis System for Axles Using Ferrous Particle Sensor”. I think the article is well organized and suitable for the "Materials" journal. But the article will be ready for publication after a major revision. Turnitin similarity rate is 13%. Comments are listed below.

1.      There is no sentence about the result in the abstract part. It should be added.

2.      The introduction should be stated in the last paragraph if the work was not done before. If there are similar studies, the difference should be clearly stated.

3.      The types of wear occurring on the axles should be mentioned in the introduction.

4.      In the experimental process section, the linear regression graphs given in Figure 7 should be shown the error rates.

5.      The chemical composition of the axle used in the experiments should be given.

6.      The study is devoid of discussions. Even if the study is new, it should be discussed by comparing it with similar studies. The literature should support it.

7.      In general, references should be increased.

8.      The article contains numerous typographic and language errors. It should be corrected.

9.      The article should be rearranged by taking into account the journal writing rules and citation rules.

10.  The paper is well-organized, yet there is a reference problem. First, your reference list contains no article from the “Materials” journal. If your work is convenient for this journal's context, then there are many references from this journal. Secondly, cited sources should be primary ones. Namely, the indexed area shows the power of a paper and directly your paper's reliability. Please make regulations in this direction.

*** Authors must consider them properly before submitting the revised manuscript. A point-by-point reply is required when the revised files are submitted.

Author Response

Dear Editor and Reviewer

We appreciate your comments.

We prepared sincere responses to your comments as follows.

The authors presented an article on “Applications and Problems of Condition Diagnosis System for Axles Using Ferrous Particle Sensor”. I think the article is well organized and suitable for the "Materials" journal. But the article will be ready for publication after a major revision. Turnitin similarity rate is 13%. Comments are listed below.

Point-1: There is no sentence about the result in the abstract part. It should be added.

“In the two tests, the first case detected a failure, but in the other case, the sensor did not detect even though a failure occurred.”

Point-2.      The introduction should be stated in the last paragraph if the work was not done before. If there are similar studies, the difference should be clearly stated.

Answer: We added sentences about the originality of our research as follows.

“In the process of developing wear particle sensors, there are studies on the sensitivity of the sensor [5, 30-33]. However, there are few researches that introduce research results using ferrous particle sensors applied to actual lubrication systems. Moreover, there is a study on the sensitivity of the sensor by the effect of debris position [34], but it is too limited research and there is no study on the location of the sensor to improve sensitivity. Therefore, we applied it to a gear system suitable for condition diagnosis using a ferrous particle sensor and tried to identify its utilization method and problems. In addition, we suggested a method for selecting the location of the sensor to improve the sensitivity of it.”

Point-3: The types of wear occurring on the axles should be mentioned in the introduction.

Answer: We mentioned about the types of wear occurring on the axles as follows.

“In axles, most of them show adhesive wear, abrasive wear and fatigue wear.”

Point-4: In the experimental process section, the linear regression graphs given in Figure 7 should be shown the error rates.

Answer: We added explanation about the error of linear regressions as follows.

“The root mean squares of the linear fitting in Figure 7 (a) and (b) were 0.983 and 0. 974, respectively.”

Point-5: The chemical composition of the axle used in the experiments should be given.

Answer: If this comment refers to the composition of the lubricant used in the axle, it is shown in Table 2. If we didn’t understand the question well, please comment again. We will then respond or make corrections.

Point-6: The study is devoid of discussions. Even if the study is new, it should be discussed by comparing it with similar studies. The literature should support it.

Answer: We have added an explanation of similar studies in the introduction and also added related references as follows.

In the process of developing wear particle sensors, there are studies on the sensitivity of the sensor [5, 30-33]. However, there are few researches that introduce research results using ferrous particle sensors applied to actual lubrication systems. Moreover, there is a study on the sensitivity of the sensor by the effect of debris position [34], but it is too limited research and there is no study on the location of the sensor to improve sensitivity. Therefore, we applied it to a gear system suitable for condition diagnosis using a ferrous particle sensor and tried to identify its utilization method and problems. In addition, we suggested a method for selecting the location of the sensor to improve the sensitivity of it.

Reference

  1. Jeon, H.G.; Kim, J.K.; Na, S.J.; Kim, M.S.; Hong, S.H. Application of condition monitoring for hydraulic oil using tuning fork sensor: a case on hydraulic system of earth moving machinery. Materials 2022, 15, 7657.
  2. Fasihi, P., Kendall, O.; Abrahams, R.; Mutton, P.; Qiu, C.; Schlafer, T.; Yan, W. Tribological properties of laser cladded alloys for repair of rail components. Materials 2022, 15, 7466.
  3. Ren. Y.; Li, W.; Zhao, G.; Feng, Z.; Inductive debris sensor using one energizing coil with multiple sensing coils for sensitivity improvement and high throughput. Tribol. Int. 2018, 128, 96-103.
  4. Xiao, H.; Wang, X.; Li, H.; Luo, J.; Fong, S. An Inductive debris sensor for large-diameter lubricating oil circuit based on a high-gradient magnetic field. Appl. Sci. 2019, 10, 1546.
  5. Ma, L.; Zhang, H.; Qiao, W.; Han, X.; Zeng, L.; Shi, H. Oil metal debris detection sensor using ferrite core and flat channel for sensitivity improvement and high throughput. IEEE Sens. J. 2020, 20, 7303-7309.
  6. Zeng, L.; Yu, Z.; Zhang, H.; Zhang, X.; Chen, H. A high sensitive multi-parameter micro sensor for detection of multi-contamination in hydraulic oil. Sens. Actuators A Phys. 2018, 282, 197-205.
  7. Ma, L.; Zhang, H.; Zheng, W.; Shi, H.; Wang, C.; Xie, Y. Investigation on the effect of debris position on the sensitivity of the inductive debris sensor. IEEE Sens. J. DOI: 10.1109/JSEN.2022.3155256.

Point-7: In general, references should be increased.

Answer: We added references as follows.

  1. Li, W.; Bai, C.; Wang, C.; Zhang, H.; Ilerioluwa, L.; Wang, X.; Yu, S.; Li, G. Design and research of inductive oil pollutant detection sensor based on high gradient magnetic field structure. Micromachines 2021, 12, 638.
  2. Feng, S.; Yang, L.; Qiu, G.; Luo, J.; Li, R.; Mao, J. An inductive debris sensor based on high-gradient magnetic field. IEEE Sens. J. 2019, 19, 2879-2886.
  3. Wu, S.; Liu, Z.; Yu, K.; Fan, Z.; Yuan, Z.; Sui, Z.; Yin, Y.; Pan, X. A novel multichannel inductive wear debris sensor based on time division multiplexing. IEEE Sens. J. 2021, 21, 11131-11139.
  4. Hong, W.; Li, T.; Wang, S.; Zhou, Z. A general framework for aliasing corrections of inductive oil debris detection based on artificial neural networks. IEEE Sens. J. 2020, 20, 10724-10732.
  5. Muthuvel, P.; George, B.; Ramadass, G.A. A highly sensitive in-line oil wear debris sensor based on passive wireless LC sensing. IEEE Sens. J. 2021, 21, 6888-6896.
  6. Muthuvel, P.; George, B.; Ramadass, G.A. Magnetic-capacitive wear debris sensor plug for condition monitoring of hydraulic systems. IEEE Sens. J. 2018, 18, 9120-9127.

  1. Jeon, H.G.; Kim, J.K.; Na, S.J.; Kim, M.S.; Hong, S.H. Application of condition monitoring for hydraulic oil using tuning fork sensor: a case on hydraulic system of earth moving machinery. Materials 2022, 15, 7657.
  2. Fasihi, P., Kendall, O.; Abrahams, R.; Mutton, P.; Qiu, C.; Schlafer, T.; Yan, W. Tribological properties of laser cladded alloys for repair of rail components. Materials 2022, 15, 7466.
  3. Ren. Y.; Li, W.; Zhao, G.; Feng, Z.; Inductive debris sensor using one energizing coil with multiple sensing coils for sensitivity improvement and high throughput. Tribol. Int. 2018, 128, 96-103.
  4. Xiao, H.; Wang, X.; Li, H.; Luo, J.; Fong, S. An Inductive debris sensor for large-diameter lubricating oil circuit based on a high-gradient magnetic field. Appl. Sci. 2019, 10, 1546.
  5. Ma, L.; Zhang, H.; Qiao, W.; Han, X.; Zeng, L.; Shi, H. Oil metal debris detection sensor using ferrite core and flat channel for sensitivity improvement and high throughput. IEEE Sens. J. 2020, 20, 7303-7309.
  6. Zeng, L.; Yu, Z.; Zhang, H.; Zhang, X.; Chen, H. A high sensitive multi-parameter micro sensor for detection of multi-contamination in hydraulic oil. Sens. Actuators A Phys. 2018, 282, 197-205.
  7. Ma, L.; Zhang, H.; Zheng, W.; Shi, H.; Wang, C.; Xie, Y. Investigation on the effect of debris position on the sensitivity of the inductive debris sensor. IEEE Sens. J. DOI: 10.1109/JSEN.2022.3155256.

Point-8: The article contains numerous typographic and language errors. It should be corrected.

Answer: We corrected typographical and language errors.

Ex) Revise Journal abbreviation : “Sens. Actuators A Phys.”, “Appl. Sci.”, “Front. Built Environ.”

Ex) refelected→reflected

Point-9: The article should be rearranged by taking into account the journal writing rules and citation rules.

Answer: We revised errors and took into account the journal writing rules and citation rules.

Point-10:  The paper is well-organized, yet there is a reference problem. First, your reference list contains no article from the “Materials” journal. If your work is convenient for this journal's context, then there are many references from this journal. Secondly, cited sources should be primary ones. Namely, the indexed area shows the power of a paper and directly your paper's reliability. Please make regulations in this direction.

Answer: We added articles from the “Materials” journal as follows.

  1. Jeon, H.G.; Kim, J.K.; Na, S.J.; Kim, M.S.; Hong, S.H. Application of condition monitoring for hydraulic oil using tuning fork sensor: a case on hydraulic system of earth moving machinery. Materials 2022, 15, 7657.
  2. Fasihi, P., Kendall, O.; Abrahams, R.; Mutton, P.; Qiu, C.; Schlafer, T.; Yan, W. Tribological properties of laser cladded alloys for repair of rail components. Materials 2022, 15, 7466.

We appreciate your comments during the first paper review process.

It is thought that the completeness of our paper has been improved more through the correction of each comment.

Thank you once again for reviewing our paper.

Sincerely yours,

Sung-Ho Hong

Round 2

Reviewer 1 Report

Authors addressed major portion of my comments in the revised manuscript. The current version of the manuscript can be accepted for publication in Materials.

Reviewer 3 Report

Journal: Materials (ISSN 1996-1944)

Manuscript ID: materials-2113709

Review Report 2#

The authors completed the requested corrections. In my opinion, this article can be accepted for publication in the "Materials" journal in its final form.